# Public Health Employees’ Perceptions about the Impact of Emerging Public Health Trends on Their Day-to-Day Work: Effects of Organizational Climate and Culture

**DOI:** 10.3390/ijerph18041703

**Published:** 2021-02-10

**Authors:** Kristie C. Waterfield, Gulzar H. Shah, Linda Kimsey, William Mase, Jingjing Yin

**Affiliations:** 1Department of Health Policy and Community Health, Jiann-Ping Hsu College of Public Health, Georgia Southern University, Statesboro, GA 30460, USA; gshah@georgiasouthern.edu (G.H.S.); lkimsey@georgiasouthern.edu (L.K.); wmase@georgiasouthern.edu (W.M.); 2Department of Biostatistics, Epidemiology, and Environmental Health Sciences, Jiann-Ping Hsu College of Public Health, Georgia Southern University, Statesboro, GA 30460, USA; jyin@georgiasouthern.edu

**Keywords:** public health workforce, workplace environment, PH WINS, organizational climate

## Abstract

*Objective*: The purpose of this research was to assess the workforce characteristics associated with public health employees’ perceived impact of emerging trends in public health on their day-to-day work. *Methods*: Multinomial logistic regression was performed to analyze data from the 2017 PH WINS, a cross-sectional survey utilizing a nationally representative sample of the United States public health workforce. *Results*: More than 55% of the public health workforce perceived that their day-to-day work was impacted by the emerging public health trends. Workplace environment was significantly associated with the perception of their day-to-day work being impacted by emerging public health trends such as quality improvement (QI) (AOR = 1.04, *p* < 0.001), and evidence-based public health practice (EBPH) (AOR = 1.04, *p* < 0.001). Race, ethnicity, and educational status were also positively associated with the perceived impact of the emerging public health trends. *Conclusions*: The organizational culture of a public health agency influences the engagement of the workforce and their perception of the meaningfulness of their work. As practitioners shift into chief health strategists, it will be imperative for them to have training in public health foundations and tools in order to efficiently serve their communities.

## 1. Introduction

The term public health evokes a variety of different images. To some, public health embodies a broad system of health professionals whose responsibility it is to solve a community’s health problems. Another image is that of a workforce that contains the knowledge, research, interventions, and techniques that can be applied to a wide variety of health-related issues and problems [1]. However, for the majority of the general public, the image of public health primarily involves governmental agencies providing medical care to indigent and underserved populations. As the practice of public health constantly evolves, the public health workforce is continually facing a barrage of challenges and obstacles that continue to cause strain on how their day-to-day practice, e.g., unexpected natural disasters, new approaches to health care, environmental emergencies, and an aging population. Because of these obstacles and challenges, public health practitioners find it difficult to remain up-to-date on the necessary knowledge and skills to effectively deliver to their communities the essential core public health services [2,3]. 

Over the last decade, the number of public health workers in the United States has been on the decline, due to high turnover rates, decreases in funding, non-competitive wages, and a large proportion of workers that are eligible for retirement [4,5,6,7,8,9]. As previous studies in organizational culture and job satisfaction have shown, many of the issues related to high-turnover and early retirement rates can be lessened when workers feel that they are supported by leadership, that their work is valued, and that they believe that the work that they are doing has value [10,11,12]. In the United States, the public health infrastructure is divided into three governmental levels: local, state, and federal. However, no single entity has complete authority and oversight of the public health infrastructure. At the federal level, Congress has the limited authority of providing commerce and allocating resources to federal agencies. The state government has the authority to provide commerce and allocate resources to its local entities, as well as, pass public health laws. The local governments have control over the local health department, as well as, oversight of the local public health ordinances and allocation of local resources [13]. As the heart of the United States public health infrastructure [3], the public health workforce takes pride in being a diverse, multidisciplinary workforce that stems from many different academic, experiential, and professional backgrounds, while ultimately sharing the common bonds of upholding the same ethical principles and being committed to the same common mission [8,14,15]. While many public health workers have received formal training that focuses solely on their specific aspect of public health practice, such as environmental health, nursing, administration, health education, or epidemiology [14], the majority lack formal training in other aspects of public health practice such as informatics, strategic planning and thinking, cultural competency, quality assurance, policy development, advocacy, community-based research, and organizational effectiveness [3,16]. Thus, there may be gaps in the skills and knowledge necessary for them to understand and apply the current emerging public health trends to their everyday practice. 

The history of public health practice in the United States has been characterized by evolution in focus, threats, and service provision. Before the 1850s it was battling recurring epidemics, and in the 1950s it focused on filling the gaps in medical care. Since 2000, the trends have focused on preparing for and responding to community health threats and providing population health services [1]. In 2015, Erwin and Brownson identified the current emerging public health trends as cross-jurisdictional sharing, creating a culture of quality improvement, Health in All Policies (HiAP), and Evidence-Based Public Health (EBPH) practice [17]. In 2016, the advancement of Public Health 3.0 began the movement that would suggest that in order to effectively improve the overall health of the population, public health practitioners need to become the chief health strategists within their communities [18]. For practitioners, knowledge regarding the emerging trends and the ability to incorporate them in the day-to-day activities of public health practice will increase efficiency through improved administrative practices, as well as, enhance proficiency in delivering the essential public health services to their communities [19,20,21,22,23,24,25,26,27]. As history has shown, the emerging trends in public health are not only related to current core competencies and skills but are also the most important future public health practice predictors in regard to program and policy implementation and success [14].

The aim of this research study seeks to answer the following questions: (1) What is the extent to which state and local public health employees in the United States perceive the impact of the six identified emerging public health issues on the day-to-day work of state and local public health workforce? (2) Is the workplace environment of public health employees associated with variations in perceived individual impact levels on the day-to-day work of state and local public health workforce? and (3) What individual characteristics of public health practitioners are associated with variations in perceived individual impact levels? This study’s focus on examining the variations in the individual perceived impact levels of the emerging issues in public health with the individual characteristics coupled with the workforce environment that may be associated with these variations will fill important gaps in the existing research literature. This study will provide a better understanding of the necessary investment in workforce development, as well as, the changes to the overall workplace environment that will allow public health workers to feel that their work is related to the overall goals of the organization [28].

## 2. Methods

This study utilized data from the 2017 Public Health Workforce Interests and Needs Survey (PH WINS). The 2017 PH WINS was conducted by the Association of State and Territorial Health Officials (ASTHO) and de Beaumont Foundation and it is the only nationally-representative survey of the United States public health workforce, at both state and local levels [29,30,31]. The aims of the survey were: to inform the public health workforce regarding future development initiatives; create a key workforce development metrics baseline; and explore the attitudes, morale, and climate of the public health workforce [29,30]. The 2017 PH WINS utilized two distinctive sampling frames, a state sampling frame, and a local sampling frame. This allowed for major considerations regarding jurisdiction population size, governing classification, and geographic location of the jurisdiction [30,31]. Additionally, the local frame only included medium and large local health departments; which meant that local health departments with a jurisdiction population smaller than 25,000 and that employ less than 25 staff members were not included [30]. The survey was fielded to staff members at participating agencies via an email invitation. The de Beaumont Foundation and ASTHO utilized an identified workforce champion and information technology contact in order to ensure the staff listings were up to date and that there would be no technical issues [30]. The survey was completed by 47,756 governmental public health workers at both the state and local levels and achieved a 48% overall response rate (state sampling frame 35% and local sampling frame 59%) [30,31]. For this analysis, we used balanced repeated replication weights for the estimates of means, frequencies, and regression coefficients. The purpose of the weights was to account for any subsampling of state-level staff, for disproportionate nonresponse by known agency staffing levels, and any poststratification nonresponse adjustments for known staffing levels at the state and national levels. Additional details about the 2017 PH WINS sampling design and size are discussed by Leider et al. [30].

### 2.1. Measures

Six dependent variables comprised the perceived impact level that six emerging issues in public health have on the day-to-day work of state and local public health workforce. Survey participants were asked, “To what extent do each of the following areas impact your day-to-day work?” The areas, grounded in previous literature, were: (1) cross-jurisdictional sharing of public health services; (2) fostering a culture of quality improvement (QI); (3) public and primary care integration; (4) evidence-based public health practice (EBPH); (5) Health in All Policies (HiAP); and (6) multi-sectoral collaboration. The responses were measured using a four-point Likert Scale, “Nothing at all”, “Not too much”, “Impact fair amount”, and “Impact great deal”. For this study, the perceived impact for each of the six areas was recoded into three categories, by combining the middle two categories of the Likert scale. “No impact” was coded as 0 [the original response category “Nothing at all” was renamed for better flow]. “Marginal impact” was coded as 1 [the original categories “Not too much” and “Impact fair amount” were combined]. “Significant impact” was coded as 2 [the original response category “Impact great deal” was renamed].

The primary *independent variable* is *workplace environment*, operationalized into a single continuous scale by summing the scores of the 17 statements (see detailed list in Bogaert et al. [32]) in which the participants were asked to rate their agreement, such as “The work I do is important”, “My training needs are assessed”, “Employees learn from one another as they do their work”, “My supervisor treats me with respect”, etc. The responses to the 17 statements were recorded on a five-point Likert Scale (1 to 5), “Strongly disagree”, “Disagree”, “Neither agree nor disagree”, “Agree”, and “Strongly agree”. Therefore, depending on the selection made by the participant, each score for each statement would range from 1 to 5, while the sum score of all of the statements within the workplace environment scale could range from 17 to 85. 

Other independent variables controlled for in the regression model include *supervisory status* of the study participant (Non-supervisor, Supervisor, Manager, Executive), *gender* (male, female, non-binary/other), *ethnicity* (Hispanic or Latino—yes, no), *race* (white, black/African American, other), *age* (≤30, 31–40, 41–50, 51–60, ≥61), *employer* (local government, state government, federal government, non-governmental), *length of tenure in public health practice* in years (0–5, 6–10, 11–15, 16–20, 21+), and *whether the employee degree is in Public Health* (public health, non–public health).

### 2.2. Analytical Methods

Descriptive statistics were computed for all dependent and independent variables. To model the categorical dependent variables (with three attributes), ordered logistic regression was initially considered. Since the assumption of proportional odds (or the parallel regression) was violated by our data, univariate regression and multinomial logistic regression were performed to compute six separate models, one for each of the emerging public health trends. These models assumed multivariate normal distributions and normally distributed error terms throughout the data. All analyses for this study were performed using SAS 9.4 [33]. Georgia Southern University’s Institutional Review Board approved and determined that this study exempt from full board review (H19312).

## 3. Results

The majority of study participants from state and local health departments and other agencies held the status of supervisor (72.26%) (Table 1). Sixty-two percent of the participants were state government employees and 78.36% identified as female. Hispanic public health workers are 12.89% of the workforce, while black public health workers make up 16.98% of the national public health workforce. The largest percentage of the workforce (30.45%) had a tenure of 5 years or less, followed by those with tenure of 21+ years (21.33%). Only 13.81% have at least one degree in public health. The mean workplace environment score (possible range of 17–85) was 66.02. 

Figure 1 presents the descriptive statistics for the perceived impact of the emerging public health issues on the day-to-day work of state and local public health workforce. The majority of public health workers perceived that they were marginally impacted during their day-to-day work by the six emerging trends: cross-jurisdictional sharing of public health services (68.54%), fostering a culture of QI (63.53%), public and primary care integration (63.51%), EBPH (58.56%), HiAP (66.85%), and multi-sectoral collaboration (66.88%). However, a significant percentage of public health workers perceived that their day-to-day work was significantly impacted by cross-jurisdictional sharing of public health services (18.18%), fostering a culture of QI (29.50%), public and primary care integration (21.02%), EBPH (29.63%), HiAP (17.23%), and multi-sectoral collaboration (22.61%). 

The results of the univariate regression models for each of the six emerging trends, showed that the workplace environment has a significant positive association with the individual perceived impact levels of all six of the emerging public health trends. Only perceptions of significant impact are discussed here. The odds of the perceiving the six emerging trends had significant impact (vs. no impact) on public health employees’ day-to-day work increased with an increase in the workplace environment score: cross-jurisdictional sharing (odds ratio [OR] = 1.02, *p*-value [*p*] 0.01), QI (OR = 1.04, *p* < 0.001), public health and primary care integration (OR = 1.02, *p* < 0.001), EBPH (OR = 1.03, *p* < 0.001), HiAP (OR = 1.03, *p* < 0.001), and multi-sectoral collaboration (OR = 1.02, *p* 0.01). 

Table 2 provides results of multinomial logistic regression models for each of the six emerging trends included in the survey. Perceptions of significant impact (vs. no impact) and perceptions of marginal impact (vs. no impact) are shown. Only perceptions of significant impact are discussed here. Both workplace environment and supervisory status of the public health worker have a significant positive association with the individual perceived impact levels of all six of the emerging public health trends. The odds of the perceiving that cross-jurisdictional sharing had significant impact (vs. no impact) on public health employees’ day-to-day work increased with an increase in the workplace environment score (adjusted odds ratio [AOR] = 1.02, *p* 0.01). Similar relationships between perceptions of importance of the remaining five trend areas and workplace environment were observed: QI (AOR = 1.04, *p* < 0.001), public health and primary care integration (AOR = 1.03, *p* < 0.001), EBPH (AOR = 1.04, *p* < 0.001), HiAP (AOR = 1.03, *p* < 0.001), and multi-sectoral collaboration (AOR = 1.02, *p* < 0.001). Being a public health agency executive (vs. non-supervisor) significantly increased the odds of having the perception that all six public health trends had a significant impact on their day-to-day work, including the impact of cross-jurisdictional sharing (AOR = 1.97, *p* 0.03), QI (AOR = 14.66, *p* < 0.001), public health and primary care integration (AOR = 1.60, *p* 0.01), EBPH (AOR = 1.90, *p* 0.04), HiAP (AOR = 2.01, *p* < 0.001), and multi-sectoral collaboration (AOR = 3.45, *p* < 0.001). Being either a manager or supervisor, for the most part, was also significantly associated with increased odds of having the perception of being significantly impacted by the emerging trends. 

Odds of having the perception of being significantly impacted in their day-to-day work when compared to not being impacted were significantly higher in minority public health workers (vs white public health workers) for at least five of the six of the emerging trends. Black public health workers had significantly increased odds in all six of the trends: cross-jurisdictional sharing (AOR = 3.17, *p* < 0.001), QI (AOR = 1.85, *p* < 0.001), public health and primary care integration (AOR = 2.80, *p* < 0.001), EBPH (AOR = 2.15, *p* 0.01), HiAP (AOR = 3.37, *p* < 0.001), and multi-sectoral collaboration (AOR = 2.41, *p* < 0.001). When compared to non-Hispanic public health workers, Hispanic workers have significantly increased odds of having the perception of being significantly impacted by cross-jurisdictional sharing (AOR = 1.81, *p* < 0.001), public health and primary care integration (AOR = 1.77, *p* < 0.001), HiAP (AOR = 1.67, *p* 0.01), and multi-sectoral collaboration (AOR = 1.56, *p* < 0.001) as opposed to not being impacted. The odds of female practitioners, as opposed to male practitioners, having the perception of being significantly impacted (vs. not impacted) by QI, public health, and primary care integration, and HiAP were significantly increased (AOR = 1.44, *p* < 0.001; AOR = 1.58, *p* 0.01; AOR = 1.67, *p* < 0.001). 

Public health workers with a public health degree (vs. those without) had significantly increased odds of having the perception of being significantly impacted as opposed to not being impacted by cross-jurisdictional sharing (AOR = 1.83, *p* < 0.001), QI (AOR = 1.69, *p* < 0.001), EBPH (AOR = 2.83, *p* < 0.001), and multi-sectoral collaboration (AOR = 3.44, *p* < 0.001). Employer type and tenure also produced some significant results. Specifically, non-governmental employer type (vs state government) had decreased odds for EBPH (AOR = 0.58, *p* 0.03). Finally, public health practitioners that were 61 years of age or older, as compared with practitioners that were 30 years old or younger, have significantly decreased odds of having the perception of being significantly impacted (vs. not being impacted) in their day-to-day work for cross-jurisdictional sharing (AOR = 0.47, *p* 0.02), public health and primary care integration (AOR = 0.53, *p* 0.01), EBPH (AOR = 0.40, *p* 0.01), and multi-sectoral collaboration (AOR = 0.41, *p* 0.03); practitioners that are 41 to 51 years of age had significantly decreased odds of having the perception of being significantly impacted by EBPH.

## 4. Discussion

In an era where modern-day public health practice is increasingly evolving to meet new demands and to address new threats, the role of the public health worker has continued to adapt to address these changes. Many public health practitioners have had to adapt their day-to-day practice in order to meet the challenges and obstacles associated with a workforce that is comprised of so many diverse backgrounds and has a majority of workers that lack formal public health training [1,34,35,36,37]. Overall, this study found that on an individual-level the workforce environment, as well as the public health worker characteristics of race, ethnicity, educational status, and supervisory status were statistically significantly associated with the perceived significant impact of each of the six identified emerging issues on the day-to-day work of state and local public health workforce. The study findings regarding the association between an emerging issue significantly impacting an individual’s day-to-day practice and the level of perceived support in the workplace environment was consistent with previous studies [38,39]. While the previous studies focused on private organizations [38] or behavioral health settings [39], our study found that the organizational climate and culture of a governmental public health agency (state or local) influences the workplace environment; which in turn can either positively or negatively impact the day-to-day engagement of the workforce and the overall perception of the meaningfulness for their day-to-day work. The association was highest for EBPH and fostering a culture of QI. This was not a surprise, since for many years the impact of EBPH and QI on health outcomes, workforce training, and organizational culture have been at the forefront of public health practice research [40,41,42].

The findings of this study demonstrate that the U.S. public health workforce is largely comprised of non-Hispanic white females that are in a supervisor role and do not have a public health degree. These findings are consistent with previous studies regarding the U.S. public health workforce composition [43]. However, black public health practitioners perceived being significantly impacted in their day-to-day work by all of the emerging public health trends when compared to white practitioners: practitioners of other races perceived being significantly impacted by five out of the six of the emerging health trends. Public health workers who identified their ethnicity as Hispanic, when compared to non-Hispanic workers, perceived significant impact of four of the six emerging health trends on their day-to-day work. The comparative impact of these findings supports the need for more diversity within the U.S. public health workforce, as well as within public health degree programs. 

Our findings also demonstrate the importance of public health workers having a public health degree. The results of this study showed that a public health degree was positively associated with individual workers perceiving significant impact of four of the six emerging issues on their day-to-day work. These results seem to suggest that mastery of the skills and knowledge associated with the emerging public health trends appear to influence the individual practitioner’s perceived impact on their daily public health practice. The association of having a public health degree with the perception that public health trends impact employees’ work may also mean that formal public health training improves employees’ ability to stay updated about changing public health environment, preparing them mentally and functionally to adopt emerging best practices such as EBPH, cross-jurisdictional sharing, fostering a culture of QI, and collaborations with community partners. Supervisory status and tenure in public health practice also appear to have an influence on a worker’s individual perception of impact of the emerging trends on their daily practice. These results may suggest that mastery of the skills and knowledge associated with the emerging public health trends not only come from formal public health training but also appear to have been learned on the job by those in senior-level supervisory roles and/or those with a long tenure in public health practice. Additionally, the results of this study may be helpful and relatable to healthcare and other non-public health organizations.

Future studies could examine the linkage between the impact of emerging public health trends and the extent to which they affect public health workforce performance and efficiency. A crucial next step for research could be investigation into the associations between individual public health practitioner characteristics and organizational capacity in terms of strengthening public health agency capability of delivering the essential public health services and improve population health outcomes. Finally, there is a need for future research that explores the association between increased opportunities for formal public health education and building public health workforce capacity at both individual and organizational levels.

### Limitations of the Study

The study has some limitations. The data was self-reported, secondary data. This exclusion of small health departments from the sampling frame could present challenges for broader generalization. An additional potential limitation is that more than 95% of those that participated in the survey were employed in supervisory roles, which could suggest that the web-based approach to fielding the survey could have hindered its availability to reach field staff and other public health workers in non-supervisory roles. Finally, this cross-sectional study design allowed only for associations to be assessed, rather than determining causal factors.

## 5. Conclusions

The practices and trends covered in this study are considered the defining features of the modern public health enterprise. The current study found that there are some gaps in if and how employees perceive an impact of these emerging trends as best practices on their public health work, including the trends such as fostering the culture of quality improvement, evidence-based public health, cross-jurisdictional sharing, health in all policies. This study also yielded actionable evidence in the form of significant variation in the perceived impact of the public health trends by employee characteristics such as having public health degree, supervisory status, race, ethnicity, and gender. These findings may provide direction as to where to target policy and capacity-building efforts for the ongoing professional development of the public health workforce. An effective public health workforce will need to continue to demand not only an increased investment from the public health agencies and their leadership, but also positive and supportive workplace environments. This study offers a sound approach for evaluating the perceived impact of the current emerging public health issues on the day-to-day work of state and local public health agency workforces and thus can be replicated over time to monitor how the emerging trends change and the effect those changes have on the public health workforce.

## Figures and Tables

**Figure 1 ijerph-18-01703-f001:**
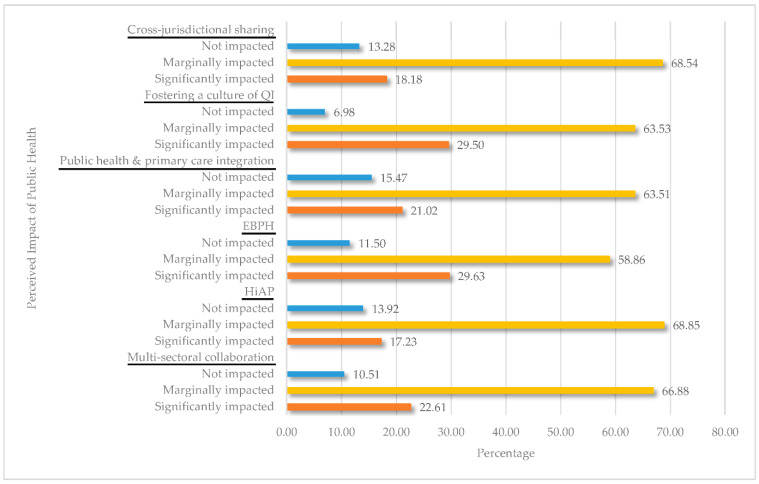
Percent distribution of employees by the perceived impact of the emerging public health issues on the day-to-day work of state and local public health workforce, 2017 Public Health Workforce Interests and Needs Survey (PH WINS). Abbreviations: QI, quality improvement; EBPH, evidence-based public health; HiAP, Health in All Policies.

**Table 1 ijerph-18-01703-t001:** Public health workforce descriptive statistics, 2017 Public Health Workforce Interests and Needs Survey (PH WINS).

	*N* (Un-Weighted)	Percent (Weighted)
Total Number of Respondents	47,756	
Supervisory Status:		
Supervisor	31,750	72.26
Manager	7017	16.39
Executive	3721	8.92
Non-supervisor	1055	2.44
Gender:		
Male	9270	21.06
Female	33,547	78.36
Non-binary/Other	301	0.58
Hispanic or Latino:		
No	36,616	87.11
Yes	6345	12.89
Race:		
White	28,410	67.37
Black or African American	6930	16.98
Other	6663	15.66
Age:		
(≤30 years)	4575	11.04
(31–40 years)	8899	22.44
(41–50 years)	10,495	24.32
(51–60 years)	12,450	28.83
(≥61 years)	5785	13.37
Employer:		
Local government	10,886	33.72
State government	31,388	62.10
Federal government	515	2.09
Non-governmental	490	2.10
Tenure in Public Health Practice:		
0–5 years	13,315	30.45
6–10 years	7458	18.41
11–15 years	6217	15.59
16–20 years	5258	14.22
21 years or above	9341	21.33
Whether the employee degree is in Public Health:		
Non-Public Health degree	37,370	86.19
Public Health degree	6329	13.81
	N	Mean (variance)
Workplace Environment	43,575	66.02 (0.17)

Abbreviations: *N*, number of observations.

**Table 2 ijerph-18-01703-t002:** Multinomial logistic regression of the perceived impact of the emerging public health trends on the day-to-day work of state and local public health workforce.

Public Health Practitioner Characteristics	Cross-Jurisdictional Sharing				Fostering a Culture of QI				Public Health & Primary Care Integration				EBPH				HIAP				Multi-Sectoral Collaboration			
	Significant Impact vs. No Impact		Marginal Impact vs. No Impact		Significant Impact vs. No Impact		Marginal Impact vs. No Impact		Significant Impact vs. No Impact		Marginal Impact vs. No Impact		Significant Impact vs. No Impact		Marginal Impact vs. No Impact		Significant Impact vs. No Impact		Marginal Impact vs. No Impact		Significant Impact vs. No Impact		Marginal Impact vs. No Impact	
	AOR	*p-Value*	AOR	*p-Value*	AOR	*p-Value*	AOR	*p-Value*	AOR	*p-Value*	AOR	*p-Value*	AOR	*p-Value*	AOR	*p-Value*	AOR	*p-Value*	AOR	*p-Value*	AOR	*p-Value*	AOR	*p-Value*	
Work Environment	**1.02**	*0.01*	**1.01**	*<0.001*	**1.04**	*<0.001*	**1.02**	*<0.001*	**1.03**	*<0.001*	**1.02**	*<0.001*	**1.04**	*<0.001*	**1.02**	*<0.001*	**1.03**	*<0.001*	**1.02**	*0.01*	**1.02**	*<0.001*	**1.02**	*<0.001*	
Supervisory Status:																									
Supervisor	0.70	*0.21*	0.75	*0.42*	**1.68**	*<0.001*	**1.50**	*<0.001*	0.72	*0.14*	0.76	*0.38*	0.72	*0.34*	0.77	*0.45*	**0.48**	*0.05*	0.73	*0.39*	0.87	*0.68*	0.83	*0.63*	
Manager	**1.47**	*0.04*	1.09	*0.53*	**4.98**	*<0.001*	**2.53**	*<0.001*	1.36	*0.16*	1.17	*0.15*	1.34	*0.09*	1.19	*0.26*	0.92	*0.64*	0.95	*0.72*	**2.90**	*<0.001*	**1.91**	*<0.001*	
Executive	**1.97**	*0.03*	1.58	*0.06*	**14.66**	*<0.001*	**5.50**	*<0.001*	**1.60**	*0.01*	**1.73**	*<0.001*	**1.90**	*0.04*	**1.61**	*0.02*	**2.01**	*<0.001*	**1.70**	*0.01*	**3.45**	*<0.001*	**1.84**	*0.01*	
Non-supervisor	--		--		--		--		--		--		--		--		--		--		--		--		
Gender:																									
Female	0.70	*0.97*	0.91	*0.55*	**1.44**	*<0.001*	**1.23**	*0.01*	**1.58**	*0.01*	**1.20**	*0.05*	1.10	*0.71*	0.91	*0.63*	**1.67**	*<0.001*	1.05	*0.77*	1.16	*0.38*	1.16	*0.10*	
Non-binary/Other	0.70	*0.48*	0.59	*0.21*	1.69	*0.25*	1.83	*0.08*	1.26	*0.61*	1.01	*0.98*	0.68	*0.47*	0.66	*0.32*	0.93	*0.92*	0.99	*0.98*	0.92	*0.88*	0.69	*0.36*	
Male	--		--		--		--		--		--		--		--		--		--		--		--		
Hispanic or Latino:																									
Yes	**1.81**	*<0.001*	1.21	*0.08*	1.06	*0.67*	0.85	*0.28*	**1.77**	*<0.001*	1.21	*0.07*	1.22	*0.05*	1.12	*0.31*	**1.67**	*0.01*	1.07	*0.57*	**1.56**	*<0.001*	1.12	*0.31*	
No	--		--		--		--		--		--		--		--		--		--		--		--		
Race:																									
Black or African American	**3.17**	*<0.001*	**1.38**	*0.01*	**1.85**	*<0.001*	0.97	*0.77*	**2.80**	*<0.001*	**1.47**	*0.01*	**2.15**	*0.01*	**1.35**	*0.01*	**3.37**	*<0.001*	**1.66**	*0.01*	**2.41**	*<0.001*	**1.24**	*0.03*	
Other	**1.66**	*<0.001*	1.20	*0.18*	1.12	*0.26*	0.97	*0.74*	**1.69**	*<0.001*	**1.35**	*0.02*	**1.34**	*0.01*	1.26	*0.09*	**2.50**	*0.01*	**1.36**	*0.01*	**1.53**	*0.01*	1.20	*0.24*	
White	--		--		--		--		--		--		--		--		--		--		--		--		
Age:																									
(31–40 years)	1.22	*0.13*	0.96	*0.66*	1.13	*0.40*	0.93	*0.70*	0.97	*0.77*	1.02	*0.85*	0.93	*0.54*	0.85	*0.18*	0.73	*0.39*	1.25	*0.07*	1.07	*0.68*	0.99	*0.97*	
(41–50 years)	1.02	*0.91*	0.77	*0.09*	1.26	*0.08*	0.79	*0.21*	1.01	*0.95*	0.85	*0.13*	**0.73**	*0.03*	**0.66**	*0.01*	0.74	*0.41*	**1.31**	*0.05*	1.13	*0.55*	1.01	*0.95*	
(51–60 years)	0.88	*0.39*	**0.67**	*0.01*	1.11	*0.37*	0.73	*0.12*	0.88	*0.38*	0.94	*0.42*	0.75	*0.08*	**0.63**	*0.01*	0.57	*0.20*	1.16	*0.34*	0.81	*0.19*	0.80	*0.08*	
(> 61 years)	**0.47**	*0.02*	0.43	*0.09*	1.06	*0.73*	0.78	*0.20*	**0.53**	*0.01*	0.60	*0.19*	**0.40**	*0.01*	0.39	*0.07*	0.32	*0.13*	0.62	*0.10*	**0.41**	*0.03*	0.47	*0.14*	
(< 30 years)	--		--		--		--		--		--		--		--		--		--		--		--		
Employer:																									
Local government	1.14	*0.30*	1.08	*0.37*	0.99	*0.88*	1.08	*0.35*	1.17	*0.25*	1.13	*0.11*	**1.47**	*0.01*	**1.40**	*0.01*	1.03	*0.70*	1.17	*0.15*	**1.30**	*0.04*	1.18	*0.13*	
Federal government	**3.00**	*0.01*	**3.50**	*0.04*	1.38	*0.57*	2.25	*0.34*	2.66	*0.11*	3.88	*0.11*	**2.60**	*0.02*	**4.97**	*0.01*	1.96	*0.14*	3.46	*0.16*	1.82	*0.14*	3.08	*0.08*	
Non-governmental	2.90	*0.40*	0.73	*0.18*	1.09	*0.93*	0.52	*0.08*	4.04	*0.18*	1.04	*0.92*	**0.58**	*0.03*	1.38	*0.61*	0.95	*0.91*	1.20	*0.47*	2.15	*0.53*	**0.62**	*0.03*	
State government	--		--		--		--		--		--		--		--		--		--		--		--		
Tenure in Public Health Practice:																									
6–10 years	1.27	*0.07*	1.31	*0.09*	**1.22**	*0.01*	1.23	*0.15*	1.26	*0.05*	**1.26**	*0.04*	**1.49**	*0.01*	**1.52**	*0.01*	1.36	*0.14*	1.23	*0.12*	1.26	*0.06*	1.22	*0.20*	
11–15 years	**1.53**	*0.01*	**1.46**	*0.02*	1.07	*0.53*	1.15	*0.28*	1.29	*0.06*	1.31	*0.07*	**1.24**	*0.02*	**1.69**	*0.02*	1.95	*0.13*	1.09	*0.31*	1.27	*0.12*	1.18	*0.33*	
16–20 years	0.87	*0.63*	0.99	*0.84*	**1.40**	*0.01*	**1.55**	*0.01*	1.08	*0.84*	1.06	*0.73*	1.15	*0.37*	1.10	*0.54*	1.23	*0.33*	0.87	*0.63*	0.71	*0.38*	0.68	*0.20*	
21 years or above	**1.73**	*0.01*	**1.71**	*0.02*	1.23	*0.06*	**1.34**	*0.01*	**1.57**	*0.01*	**1.46**	*0.04*	**1.62**	*0.01*	1.61	*0.09*	1.93	*0.06*	**1.40**	*0.03*	1.30	*0.09*	1.32	*0.16*	
0–5 years	--		--		--		--		--		--		--		--		--		--		--		--		
Whether the employee’s degree is in Public Health:																									
Public Health degree	**1.83**	*<0.001*	**1.23**	*0.04*	**1.69**	*<0.001*	**1.37**	*0.01*	0.99	*0.96*	0.95	*0.58*	**2.83**	*<0.001*	**1.50**	*0.03*	1.20	*0.12*	1.13	*0.24*	**3.44**	*<0.001*	**1.70**	*<0.001*	
Non-Public Health degree	--		--		--		--		--		--		--		--		--		--		--		--		

Abbreviations: QI, quality improvement; EBPH, evidence-based public health; HiAP, Health in All Policies; AOR, adjusted odds ratio; CI, confidence intervals. Note: Bold AORs indicates statistically significant differences compared with the reference category at *p* < 0.05.

## Data Availability

Publicly available dataset was analyzed in this study. This data can be found here: https://www.debeaumont.org/programs/ph-wins/.

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
