# Peer review of "Public Health Employees’ Perceptions about the Impact of Emerging Public Health Trends on Their Day-to-Day Work: Effects of Organizational Climate and Culture"

_ijerph, 2021, doi:10.3390/ijerph18041703_

Round 1

Reviewer 1 Report

Although the authors' findings are important and perhaps new in health care, all of this has long been well known in management. Therefore, the authors should complement the literature review by paying more attention to the effects of organizational culture on loyalty and job satisfaction.

The Introduction chapter is followed by Methods, and the literature review is exhausted as described in the introduction. A new chapter, Literature background, should be inserted between these two chapters, research questions should be formulated at the end, and these should be derived from this literature analysis by logical reasoning.

The Discussion chapter also needs to be revised. Although the authors' findings are correct, there is a lack of demonstration of how the relationships between organizational culture and engagement are shaped by research findings in other (non-health) organizations.

Reviewer 2 Report

The author needs to elaborate on the primary data collection procedure. The authors have mentioned that this study is secondary data analysis. However, details of the primary study, how participants were approached, and how data was collected in the 2017 public health workforce survey. Some context of the primary data has been mentioned as a limitation (312-314). However, I think it should be mentioned as the study settings rather than as a limitation. 

The authors have mentioned that the response rate was 48% (line 110). I checked with the primary study produced from the 2017  PH WINS survey ( cited as reference #27 by the authors). It is mentioned that the response rate in the survey was 59% (abstract of the paper, reference #27). The reason for this discrepancy needs to be justified. 

It is not clear whether the authors have taken permission for the further analysis of the survey conducted by a third party. 

The introduction is very long. The authors need to consider reducing the length of the introduction section. It may be compensated with addition to the methodology section. 

Reviewer 3 Report

Public Health Employees’ Perceptions about Impact of Emerg- ing Public Health Trends on their Day-to-Day Work: Effects of Organizational Climate and Culture

Research necessity and purpose

- With regard to the research necessity, there is an absence of specific description of knowledge and techniques required for public health services of communities.

- Line 82: (1) What do the six public health issues mean? Describe all the issues in detail.

- Line 85: (2-3) The researcher suggests environmental health, nursing, administration, health education, epidemiology and so on in relation to personal effects, but concrete hypotheses regarding individual characteristics and the degrees of influences should be presented.

Research method and procedure

-Line- 99: In terms of the use of PH WINS data, you have to add an explanation of where and how the data were collected.

-Line- 120: What is the basis for the six issues regarding public health employees? There is a need for a deeper explanation of where it was used previously and the basis for its composition. Explain based on the existing studies.

-Line- 138: Explain a specific way of calculating scores. Also, is the score calculation method based on Gloud’s research? What does the score calculated mean? I consider an explanation of this is needed.

- Line- 146: What is the degree of normal distribution of the data used in this study? This should be suggested additionally.

- It is required to add reliability and validity of each study scale for a clear explanation of the scales used in this study.

- Also, an accurate research ethics number(IRB) should be specified.

Research result and discussion

- <Table 1> This study classifies age in a different way from PH WINS. It seems a related explanation should be added. Also, an additional description of the standard for classification of races that the researcher set is required.

- This study says black public health employees were influenced by five out of six factors and Hispanic ones were influenced by four of six factors. Do you think this research result just demonstrates there is a need to acquire a degree in public health and that diversity should be guaranteed? You have to explain why this impact was found in different races and what contribution this result can make academically, based on the previous studies.

Round 2

Reviewer 1 Report

Thank you for your revision.

Author Response

You are very welcome. 

Reviewer 2 Report

Thank you for addressing the comments and clarification. 

The clarification for point number 2 (prevalence of 48% needs to be clarified in the text - for local level it was 59% - so that it will give readers more confidence of the results and not remove their confusion about results)

Author Response

The text in question can be found on lines 118-120.  The original text was:

The survey was completed by 47,756 public health workers and achieved a 48% response rate [30,31].

While utilizing track changes, the text in lines 118-121 has been updated to the following:

The survey was completed by 47,756 governmental public health workers at both the state and local levels and achieved a 48% overall response rate (state sampling frame 35% and local sampling frame 59%) [30,31].